# Fraction of plasma exomeres and low-density lipoprotein cholesterol as a predictor of fatal outcome of COVID-19

Tatiana Usenko [1,2,3]*, Valentina Miroshnikova[1,2], Anastasia Bezrukova[1], Katerina Basharova[1], Sergey Landa [1,2], Zoia Korobova[4], Natalia Liubimova[4], Ivan Vlasov[5], Mikhael Nikolaev[1,2], Artem Izyumchenko[1], Elena Gavrilova[2], Irina Shlyk[2], Elena Chernitskaya[2], Yurii Kovalchuk[2], Petr Slominsky[5], Areg Totolian[4], Yurii Polushin[2], Sofya Pchelina[1,2,3]

1 Petersburg Nuclear Physics Institute Named by B.P. Konstantinov of National Research Centre «Kurchatov Institute», Gatchina, Russia, 2 Pavlov First Saint-Petersburg State Medical University, Saint-Petersburg, Russia, 3 Kurchatov Genome Center—PNPI, Saint-Petersburg, Russia, 4 Saint Petersburg Pasteur Institute, Saint-Petersburg, Russia, 5 Institute of Molecular Genetics, National Research Center "Kurchatov Institute", Moscow, Russia

* tatiana.s.usenko@gmail.com

**Data Availability Statement:** All relevant data are within the paper and its Supporting information files.

## Abstract

Transcriptomic analysis conducted by us previously revealed upregulation of genes involved in low-density lipoprotein particle receptor (LDLR) activity pathway in lethal COVID-19 caused by SARS-CoV-2 virus (severe acute respiratory syndrome coronavirus 2). Last data suggested the possible role of extracellular vesicles in COVID-19 pathogenesis. The aim of the present study was to retrospectively evaluate parameters of cholesterol metabolism and newly identified EVs, exomeres, as possible predictors of fatal outcome of COVID-19 patients infected by the Alpha and the Delta variants of SARS-CoV-2 virus. Blood from 67 patients with severe COVID-19 were collected at the time of admission to the intensive care unit (ICU) and 7 days after admission to the ICU. After 30 days patients were divided into two subgroups according to outcome—34 non-survivors and 33 survivors. This study demonstrated that plasma low- and high-density lipoprotein cholesterol levels (LDL-C and HDL-C) were decreased in non-survivors compared to controls at the time of admission to the ICU. The conjoint fraction of exomeres and LDL particles measured by dynamic light scattering (DLS) was decreased in non-survivors infected by the Alpha and the Delta variants compared to survivors at the time of admission to the ICU. We first showed that reduction of exomeres fraction may be critical in fatal outcome of COVID-19.

## Introduction

Coronavirus infection (COVID-19) is caused by SARS-CoV-2 virus (severe acute respiratory syndrome coronavirus 2) and significant number of patients with COVID-19 develops critical illness with fatal disease outcome [1]. Despite of the numerous studies devoted to the study of the pathogenesis of COVID-19, there are still many questions, in particular, what factors may

**Funding:** This work was supported by the Genome Research Centre development program «Kurchatov Genome Centre» (agreement No.075-15-2019-1663). The funder had no role in the study design, data collection and analysis, decision to publish, or preparation of the manuscript.

**Competing interests:** The authors have declared that no competing interests exist.

determine the lethal outcome. Last data revealed the link between cholesterol metabolism and susceptibility and severity of COVID-19 [2, 3]. In particular, serum lipid levels decrease in patients with COVID-19 and low level of serum low-density lipoprotein cholesterol (LDL-C) have been proposed to be a predictor of poor disease prognosis [2, 4]. In our previous study transcriptome analysis of peripheral blood mononuclear cells (PBMCs) of patients with COVID-19 upon admission to the ICU revealed an activation of the low-density lipoprotein particle receptor (LDLR) activity pathway (GO:0005041) in deceased patients when compared with survivors [5]. Our results point out this hyperactivation of LDLR activity pathway could be linked to cholesterol metabolism disturbances expressed in changes of lipid spectrum observed in another studies. According to the latest point of view extracellular vesicles (EVs) could be considered as alternative cholesterol vehicles in blood plasma [6, 7]. We and others demonstrated the variety in the size of EVs in non-survivors compared to patients with positive outcome, that is in accordance with other studies showing the heterogeneous nature of EVs from COVID-19 patients [8–10]. EVs include large microvesicles (200–1000 nm), small exosomes (50–200 nm), and newly identified exomeres (<50 nm) that was shown to be very different from exosomes in their lipid composition [11, 12]. We suppose that exomeres may play a role in COVID-19 pathogenesis as an essential role of exomeres in cholesterol transport was shown [13]. To further analyze the role of cholesterol metabolism in poor prognosis of COVID-19 we evaluate retrospectively the parameters of cholesterol metabolism (plasma lipids concentrations and exomere fraction, the expression levels of LDLR activity pathway genes selected through our previous transcriptome analysis) as well as cytokine profile as possible predictors of fatal outcome in patients with severe COVID-19 infected by the Alpha and the Delta variants of SARS-CoV-2 virus.

## Materials and methods

### Ethics statement

The study involving human participants was approved by the Ethics Committee of the Pavlov First State Medical University of St. Petersburg (Russia). A formal written consent form was provided to all included subjects to read and sign prior to the study at the time of admission to intensive care (ICU).

### Subjects

The current study included 67 patients with severe COVID-19 infected with the Alpha and the Delta variants of SARS-CoV-2 who were admitted to the to the intensive care unit (ICU) of Pavlov First Saint-Petersburg State Medical University during the Alpha variant predominant period (between 1st November 2020 to 25th February 2021) and the Delta variant predominant period (between 1st July 2021 and 31th August 2021), respectively. Biological samples (plasma and PBMCs) were collected at the time to admission to the ICU and one week (7 days) after admission to the ICU. Depending on the disease outcome during the time of observation (30 days) patients were divided into subgroups depending of outcome: 33 survivors and 34 non-survivors. Also, all 67 COVID-10 patients were separated in accordance by variant of COVID-19: 28 COVID-19 patients infected by the Alpha variant of SARS-CoV-2: 13 non-survivors and 15 survivors and 39 COVID-19 patients infected by the Delta variant of SARS-CoV-2: 21 non-survivors and 18 survivors. The inclusion criteria were Russian ethnicity, age between 40 and 80 years with the absence of chronic comorbidities such as cancer, cerebrovascular diseases, heart failure, or renal failure but we did not exclude obesity or hypertension, as the common features in patients among patients with severed COVID-19. All patients admitted to the intensive care unit and met the inclusion criteria were included in the study. No one

**Table 1. Characteristics of the studied groups.**

| | All patients | | | Patients with COVID-19 infected by the Alpha variant | | | Patients with COVID-19 infected by the Delta variant | | | Controls (N = 40) |
|---|---|---|---|---|---|---|---|---|---|---|
| | Non-survivors with COVID-19 (N = 34) | Survivors with COVID-19 (N = 33) | p[a] | Non-survivors with COVID-19 (N = 21) | Survivors with COVID-19 (N = 18) | p[a] | Non-survivors with COVID-19 (N = 13) | Survivors with COVID-19 (N = 15) | p[a] | |
| Mean age ± SE, years | 60.45±9.8 | 55.66±10.76 | >0.05 | 60.6±10.9 | 57.6±9.1 | >0.05 | 60.1±9.3 | 55.3±12.9 | >0.05 | 62.25±7.9 |
| Sex (Male: Female) | 20:14 | 20:13 | >0.05 | 10:3 | 9:6 | >0.05 | 10:11 | 11:7 | >0.05 | 23:17 |
| Oxygen saturation * | 84.60±11.20 | 88.70±5.89 | >0.05 | 88.10±8.79 | 91.5±4.70 | >0.05 | 81.00±13.61 | 85.89±7.08 | >0.05 | - |
| T ˚C * | 37.20±0.55 | 37.30±0.63 | >0.05 | 37.00±0.24 | 37.0±0.57 | >0.05 | 37.30±0.86 | 37.50±0.68 | >0.05 | - |
| CT-SKAN * | 3–4 | 3–4 | >0.05 | 3–4 | 3–4 | >0.05 | 3–4 | 3–4 | >0.05 | - |
| Hypertension (yes:no) | 23:11 | 13:20 | <0.05 | 7:6 | 8:7 | <0.05 | 16:5 | 5:13 | <0.05 | - |
| Obesity (yes:no) | 15:19 | 3:30 | <0.05 | 4:9 | 1:14 | <0.05 | 11:10 | 2:16 | <0.05 | - |
| Neutrophils, $\times10^9$/l | 6.20 (2.41–12.30) | 9.57 (4.03–16.40) | >0.05 | 6.26 (2.41–12.30) | 9.57 (4.03–16.40) | >0.05 | 6.16 (5.31–10.33) | 11.05 (6.62–15.36) | >0.05 | - |
| Lymphocytes, $\times10^9$/l | 0.60 (0.30–1.40) | 0.90 (0.40–1.70) | <0.05 | 0.60 (0.30–1.40) | 0.85 (0.40–1.70) | <0.05 | 0.60 (0.30–0.80) | 0.95 (0.60–1.20) | <0.05 | - |
| Monocytes, $\times10^9$/l | 0.28 (0.11–0.91) | 0.44 (0.14–1.30) | >0.05 | 0.27 (0.11–0.91) | 0.42 (0.14–1.30) | >0.05 | 0.29 (0.12–0.44) | 0.55 (0.28–0.64) | >0.05 | - |
| D-dimer, µg/l | 1141 (273–6692) | 914 (287–35200) | >0.05 | 1133 (273–6692) | 950 (287–35200) | >0.05 | 1141 (448–2510) | 695 (357–1178) | >0.05 | - |
| C-reactive protein, mg/l | 141 (15–278) | 148 (5–227) | >0.05 | 154 (15–278) | 156 (5–227) | >0.05 | 138.5 (79.5–202.4) | 142.6 (103.4–191.5) | >0.05 | - |

* at the time of admission to the ICU, a–analysis of variants (ANOVA), CT-SKAN scores depending on the extent of consolidation or ground-glass opacities: 1—<25%; 2–25–50%; 3–50–75%; 4—>75%.

received the lipid-lowering therapy before and during the study. The control group included 40 healthy individuals who had never been infected with SARS-CoV-2 that was additionally confirmed by negative PCR and IgG antibody tests. The characteristics of studies groups are presented in Table 1.

## Blood plasma collection

Blood plasma was isolated from peripheral venous blood collected in vacuum containers with EDTA by centrifugation for 20 min, 3000 g. Centrifugation was carried out no later than 20 min after blood taking. Plasma samples were stored at -80˚C until the study.

## Plasma lipid profile

Total cholesterol (TC), high-density lipoprotein cholesterol (HDL), low-density lipoprotein cholesterol (LDL), triglycerides (TG) in plasma were determined by OLYMPUS AU400 apparatus.

## Plasma cytokine profile

The concentration of cytokines in blood plasma was assessed by multiplex analysis using a MagPix analyzer (Luminex Corporation, USA) using a kit for simultaneous detection of 25 analytes (IL-17F, GM-CSF, IFNγ, IL-10, CCL20/MIP3α, IL-12(p70), IL-13, IL-15, IL-17A, IL-

22, IL-9, IL-1β, IL-33, IL-2, IL-21, IL-4, IL-23, IL -5, Il-6, IL-17E/IL-25, IL-27, IL-31, TNFα, TNFβ, IL-28A) (Milliplex MAP Human Th17 Magnetic Premix 25 Plex Kit (HT17MG-14K-PX-25, Merk-Millipore, USA) according to the manufacturer's conditions. The results were processed using xPONENT 4.2 software.

## The method of dynamic light scattering (DLS) for detection of exomeres

Relative levels of exomeres and LDL particles were assessed in plasma by means of a combination of immunoadsorption and dynamic light scattering (DLS) methods as described earlier [13, 14]. This approach allows to distinguish particles by size (using hydrodynamic radius) including after preliminary depletion of unnecessary particles by immunoprecipitation with appropriate antibodies to clean particles of interest. Plasma was processed separately with CD9, HSP90 and ApoB-100 anti-bodies with subsequent separation of corresponding particles via centrifugation of 1,000 g for 10 min. Supernatants and initial plasma were analyzed by DLS. Principle of the method is represented in S1 Fig. Conjoint fraction of exomeres and LDLs particles is determined by analysis of corresponding peak from measurement of unprocessed plasma (S1A Fig). Relative level of LDL particles is determined as a mean from analysis of resulting peak after CD9+ and HSP90+ particles (exosomes and exomeres) were removed (S1B and S1C Fig). Relative level of exomeres was determined by analysis of resulting peak after ApoB-100 containing particles were removed (S1D Fig). The measurements were carried out using a laser correlation spectrometer DLS (INTOX MED LLC, St. Petersburg, Russia) with a heterogeneous measurement scheme [15]. Mathematical processing of the obtained data was carried out using the algorithm [15] using the QELSspec (version 3.4) software package, Gatchina, Russia.

## Isolation of peripheral blood mononuclear cells

PBMCs were isolated from 8 mL EDTA-anticoagulated venous blood by a Ficoll-Paque gradient method (Ficoll-Paque PLUS, GE Healthcare, Chicago, IL, USA) [16]. After centrifugation, PBMCs were collected from the interface and washed twice with PBS (pH 7.4) to remove the platelet-rich plasma fraction. The PBMCs cell pellets were aliquoted and immediately frozen at −80˚C.

## RNA extraction, cDNA synthesis and quantitative real-time PCR in PBMCs

Total RNA was extracted from PBMCs using GeneJET RNA Purification Kit (K0731, Thermo Fisher Scientific, USA) and complementary DNA (cDNA) was synthesized by Revert Aid First cDNA Synthesis kit (K1622, Thermo Fisher Scientific, USA. Primers and probes were designed by Primer3 program (https://bioinfo.ut.ee/primer3-0.4.0/) (S1 Table). The expression level of cholesterol metabolism genes (*LDLR, PPARG, LRP6, CD36, ANXA2, STAB1*) was assessed with TaqMan probes by real-time quantitative PCR on CFX96 (Bio-Rad, USA). mRNA levels of the studied genes were normalized to mRNA levels of *RPLP0* (Ribosomal Protein Lateral Stalk Subunit P0) and ACTB (beta actin). Each sample was tested in triplicate. The value of the expression level relative to the calibrator was determined by the formula: $2^{-\Delta\Delta C}$ [17].

## Droplet digital PCR (ddPCR) assay

Nucleic acid copy counts of *STAB1* were determined by means of the *RPLP0* gene as housekeeping droplets using droplet digital PCR technology (Bio-Rad, USA) following the

manufacturer´s protocol and using ddPCR EvaGreen Supermix (Bio-Rad, USA). The thermal cycling conditions for *STAB1* was standard as describe in the manufacturer´s protocol of ddPCR EvaGreen Supermix (Bio-Rad, USA) with annealing temperature 57˚C. The ddPCR reaction with primers listed above was performed in a T100 Thermal Cycler. Measurement of positive droplets per µl sample was performed on a QX200 ddPCR Droplet Reader (Bio-Rad, USA). Based on the droplet counts and according to Poisson distribution, absolute nucleic acid copy counts were calculated utilizing the software QuantaSoft (Bio-Rad, USA).

## Statistical analysis

Conformity of findings to normal distribution was tested using the Shapiro–Wilk test. Comparison of variation between the studied groups was performed by Kruskal-Wallace test. Comparison of variation between two groups was conducted using the nonparametric Mann-Whitney u-test. Correlation analysis was conducted with Spearman coefficient. Linear regression analysis was used to test for association between outcome of COVID-19 and studied parameters, adjusting for age, sex, obesity and hypertension. Significance was established at $p < 0.05$. Statistical analysis was performed using R software (version 4.1.2). Clinical data expressed as ± standard deviation (SD), experimental data as the median (min–max).

## Results

This study included the analysis of plasma lipid profile measured by standard technics and DLS. All patients admitted to the ICU were characterized with respiratory failure and lung injury with CT-SKAN scores equal to 3–4. The demographic, clinical and laboratory characteristics of the studied populations are shown in Table 1. COVID-19 patients were characterized with lymphopenia and the elevated values of neutrophils, C-reactive protein (CRP) and D-dimer that is in accordance with previous data [18]. There were no differences between survivors and non-survivors except more pronounced lymphopenia in non-survivors (Table 1). Also, patients with COVID-19 were characterized by increased secretion of proinflammatory cytokines CCL20/MIP3α, IL-10, IL-15, IL-27 in non-survivors than in controls (S1 File).

### Plasma lipid profile for all patients with various outcomes of COVID-19 (survivors / non-survivors) at the time and 7 days after admission to the ICU and in the control group

Lipid profile was estimated in plasma of all patients with COVID-19 with different outcomes of COVID-19 (survivors / non-survivors) included in the current study at the time and 7 days after admission to the ICU and in the control group by standard technics (Table 2).

At the time of admission to the ICU we showed decreased TC and LDL levels in non-survivors compared to survivors and control group (TC: p = 0.014, p<0.0001; LDL: p = 0.011, p<0.0001; respectively) and decreased TC and LDL levels in survivors compared to the control group (p = 0.012, p = 0.028, respectively). HDL level was decreased in non-survivors and survivors when comparing to the control group (p<0.0001, p = 0.00064, respectively). No statistically significant differences in TG concentration between studied groups were found (p>0.05) at the time of admission to the ICU. 7 days after of admission to the ICU, TC, LDL and HDL levels were decreased in non-survivors compared to survivors and controls (TC: p = 0.00034, p<0.0001; LDL: p = 0.0002, p<0.0001; HDL: p = 0.018, p<0.0001; respectively). At the same

**Table 2. Plasma lipid profile for patients with COVID-19 and controls.**

| Parameters | Lipid spectrum of blood plasma, median (min-max), mmol/l | | | | | | | | | | | | |
| --- | --- | --- | --- | --- | --- | --- | --- | --- | --- | --- | --- | --- | --- |
| | All patients with COVID-19 | | | | Patients with COVID-19 infected by the Alpha variant | | | | Patients with COVID-19 infected by the Delta variant | | | | Control group (N = 40) |
| | Admission to the ICU | | 7 days after admission to the ICU | | Admission to the ICU | | 7 days after admission to the ICU | | Admission to the ICU | | 7 days after admission to the ICU | | |
| | Non-survivors (N = 34) | Survivors (N = 33) | Non-survivors (N = 34) | Survivors (N = 33) | Non-survivors (N = 21) | Survivors (N = 18) | Non-survivors (N = 21) | Survivors (N = 18) | Non-survivors (N = 13) | Survivors (N = 15) | Non-survivors (N = 13) | Survivors (N = 15) | |
| TC | 3.35(2.10–5.70) **p = 1.8e-06\*** **p = 0.014\*\*\*\*** | 4.05(2.50–6.70) **p = 0.012\*** | 3.40(1.50–5.10) **p = 5.3e-05\*** **= 0.00024\*\*** | 5.00(3.10–8.93) **p = 0.018\*\*\*\*** | 3.40(2.40–4.80) **p = 0.00028\*** **p = 0.054\*\*\*\*** | 4.00(2.60–5.93) | 4.05 (2.00–4.70) **p = 0.025\*** **p = 0.018\*\*** | 5.00 (3.10–8.93) | 3.50(2.10–5.79) **p = 0.0063\*** | 4.19(2.50–6.70) | 2.80(1.5–5.10) **p = 0.00088\* = 0.0035\*\*** | 5.00(3.9–7.6) **p = 0.037\*\*\*\*** | 5.14 (2.50–7.91) |
| LDL | 1.61(0.14–3.52) **p = 3.1e-06\*** **p = 0.011\*\*\*\*** | 2.35(0.92–4.32) **p = 0.028\*** | 1.48 (0.40–3.14) **p = 2.4e-05\*** **= 0.0002\*\*** | 3.08(1.55–4.59) | 1.79(0.97–2.86) **p = 0.002\*** **p = 0.087\*\*\*\*** | 2.38(1.45–3.72) | 2.18(0.70–2.76) **p = 0.011\*** **p = 0.011\*\*** | 3.20(1.55–4.59) | 1.81(0.14–3.52) **p = 0.0013\*** | 2.22(0.92–4.32) **p = 0.062\*** | 1.36(0.21–1.18) **p = 0.0011\* = 0.0043\*\* 0.0014\*\*\*\*** | 2.68(1.80–4.681) | 3.19 (0.71–5.54) |
| HDL | 0.82(0.27–1.61) **p = 9.5e-06\*** | 0.91(0.43–1.46) **p = 0.00064\*** | 0.75(0.21–1.18) **p = 2.3e-06\* = 0.018\*\*** | 0.99(0.61–1.30) **p = 0.0018\*** | 0.76(0.40–1.61) **p = 0.00023\*** | 0.96(0.43–1.41) **p = 0.0038\*** | 0.75(0.67–1.08) **p = 0.00054\*** | 0.81(0.65–1.19) **p = 0.00082\*** | 0.83(0.27–1.39) **p = 0.018\*** | 0.89(0.56–1.46) **p = 0.067\*** | 0.63(0.21–1.18) **p = 0.001\* = 0.0067\*\* p = 0.049\*\*\*** | 1.06(0.61–1.30) | 1.15 (0.68–2.69) |
| TG | 1.56(0.88–5.42) | 1.60(0.89–3.40) | 2.18(1.19–3.40) **p = 0.0039\* = 0.042\*\*\*** | 2.65(1.02–9.66) **p = 0.00086\* = 0.0054\*\*\*\*** | 1.52(0.98–5.42) | 1.45(0.95–3.01) | 2.12(1.19–2.92) **p = 0.071\*** | 2.12(1.31–5.14) **p = 0.061\*** | 2.09(0.88–4.28) | 1.66(0.89–4.50) | 2.62(1.41–3.40) | 2.94(1.02–9.66) | 1.58 (0.54–4.23) |

\*—compared with controls,

\*\*—compared with survivors with COVID-19 (in a week),

\*\*\*—compared with non-survivors with COVID-19 (admission),

\*\*\*\*—compared with survivors with COVID-19 (admission)

time decreased level of HDL was observed in survivors compared to the control group 7 days after of admission to the ICU (p = 0.0018). TG level was increased in non-survivors and survivors compared to the control group after 7 days of admission to the ICU (p = 0.0039, p = 0.00086, respectively). TC and TG levels were increased in survivors (p = 0.018, p = 0.0054, respectively) and TG in non-survivors at 7 days after of admission to the ICU compared to the time of admission to the ICU (p = 0.042).

Also, plasma lipid profiles were compared in patients with COVID-19 depending on the variant of the SARS-CoV-2 virus (the Alpha, the Delta) (Table 2). TC, LDL and HDL cholesterol in the plasma were decreased in non-survived patients with COVID-19 infected by the Alpha variant compared to controls (p = 0.00028, p = 0.002, p = 0.00023. respectively) and TC and LDL had a tendency to decrease compared with survived patients with COVID-19 (p = 0.054, p = 0.087, respectively) at the time to admission to the ICU. Also, we demonstrated a decrease of plasma HDL cholesterol in survived patients with COVID-19 when compared with controls (p = 0.0038). No statistically significant differences in TG concentration between studied groups were found (p>0.05). In order to assess the dynamic change in plasma lipid profile of survivors and non-survivors during severe COVID-19 we additionally evaluated lipid levels in 7 days after admission to the ICU (Table 2). 7 days after admission to the ICU TC and LDL concentration were decreased in non-survivors compared with controls (p = 0.025, 0.011, respectively) and survivors (p = 0.018, p = 0.011, respectively). HDL concentration was decreased and TG concentration had tendency to increase in non-survivors and survivors compared with controls (p = 0.00054, p = 0.071; p = 0.00082, p = 0.061, respectively). A pairwise analysis comparing plasma lipid profile change in non-survivors and survivors between two observation points (the time and 7 days after admission to the ICU) did not reveal significant differences (p>0.05).

With regard to changes in the lipid profile in patients infected by the Delta variant, we also showed a decreased level of LDL and HDL cholesterol in the plasma of non-survived patients with COVID-19 when compared with controls (p = 0.0013, p = 0.018, p = 0.0021, respectively) and tendency to decrease of LDL, HDL and TC levels in survived patients with COVID-19 (p = 0.062, p = 0.067, p = 0.063, respectively). No statistically significant differences in TG concentration between studied groups were found (p>0.05).

7 days after admission HDL and LDL cholesterol as well as TC levels were decreased in non-survivors compared with survivors (p = 0.0043, p = 0.0067, p = 0.0035 respectively) and the control group. (p = 0.001, p = 0.0011, p = 0.00088, respectively) (Table 2). A pairwise analysis comparing plasma lipid profile change in non-survivors and survivors between two observation points was performed. TC concentration was increased in survivors in 7 days after admission to the ICU when compared with the time of admission (p = 0.037). At the same time non-survivors were characterized by continuing decrease of HDL and LDL cholesterol levels in 7 days after admission to the ICU when compared with the time of admission (p = 0.049, p = 0.0014, respectively). Plasma TG concentration did not differ between the studied groups neither at the time of admission nor 7 days after (p>0.05).

Overall, deceased patients with COVID-19, regardless of the variant with which they were infected (the Alpha, the Delta), were characterized by decreased LDL level at the time of admission. However, the Delta variant was associated with continuous decrease in LDL and HDL levels in deceased patients 7 days after admission in ICU. Thus, more pronounce cholesterol metabolism disturbances in patients affected with the Delta variant could be supposed. Therefore, our further study was focused on the evaluation of LDL-related parameters (exomeres, analysis of gene expression involved in LDLR activity pathway revealed in our previous study) in patients with COVID-19 with the Delta variant [5].

### Relative levels of plasma exomeres and LDL particles detected by means of DLS for patients with various outcomes of COVID-19 (survivors / non-survivors) at the time and 7 days after admission to the ICU and in the control group

Exomeres are a class of extracellular vesicles enriched in cholesterol and comparable in size with LDLs. Our approach allowed to distinguish particles by size (using hydrodynamic radius). Relative levels of exomeres and LDLs as well as conjoint fraction of exomeres and LDL particles were assessed by combination of immunoadsorption and DLS. We analyzed conjoint level of particles with hydrodynamic radius of 20 nm (exomers and LDLs), next we analyzed the same peak after immunoadsorption of exomeres using antibodies against CD9 and HSP90 (LDL particles remain after cleaning) and antibodies against apolipoproteins B100 (exomeres remain after cleaning). It should be noted that relative LDLs level determined by this method as relatively real quantity of particles is highly positively correlated with those measured by standard technique.

Level of plasma exomeres was decreased in non-survivors infected by the Delta variant compared with survivors at time to admission to the ICU (p = 0.011), still there were not any differences between the studied groups in 7 days after admission (Fig 1, S2 Table). Level of LDL particles was decreased in non-survivors at the time of admission to the UCI compared

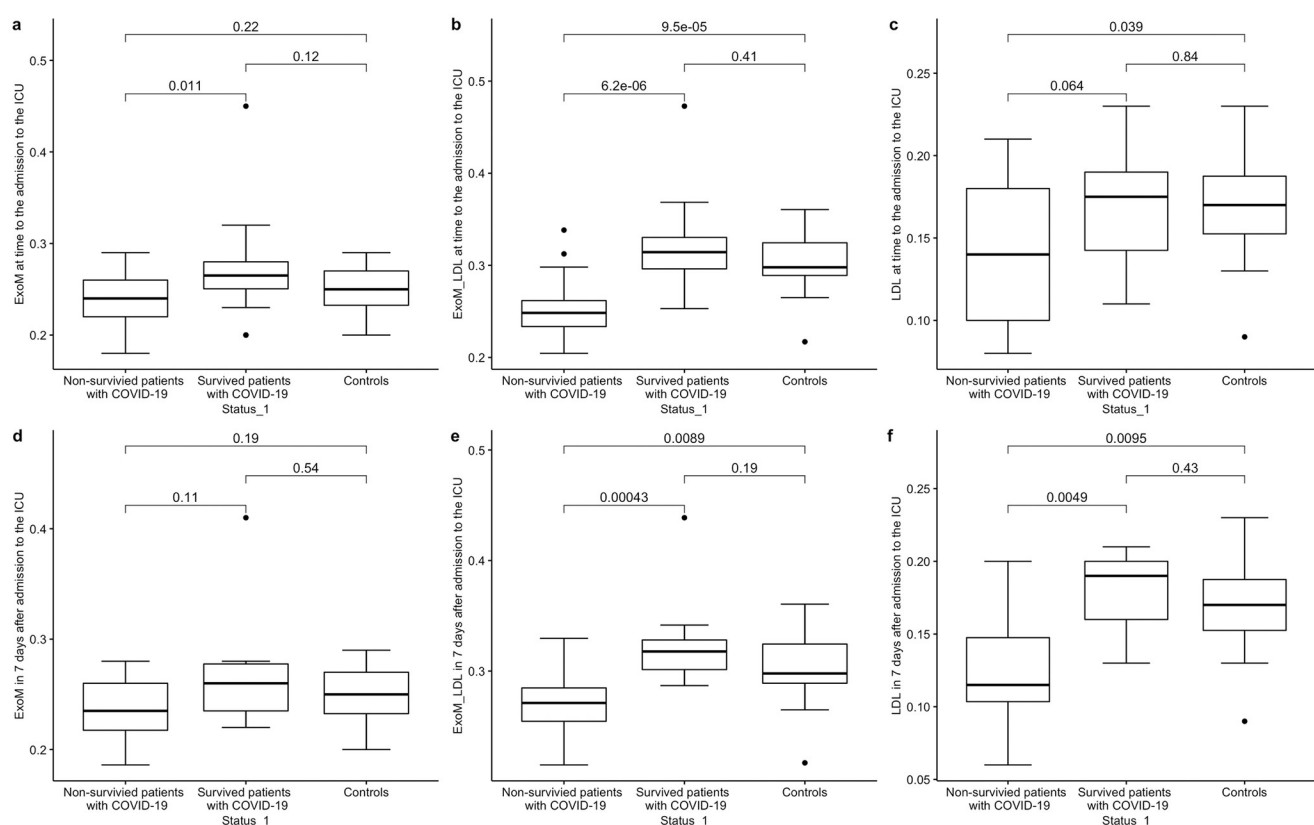

**Fig 1.** Relative level of plasma exomeres in patients with the Delta variant and controls with concentration of TC bound by LDL: a. ExoM (admission to the ICU), b. ExoM_LDL (admission to the ICU), c. LDL (admission to the ICU), d. ExoM (7 days after admission to the ICU), e. ExoM_LDL (7 days after admission to the ICU), f. LDL (7 days after admission to the ICU).

with controls (p = 0.039). 7 days after to admission to the ICU level of LDL particles was lower in non-survivors compared with survivors (p = 0.0049) and controls (p = 0.0095). Interestingly, if one takes into account conjoint fraction of exomeres and LDL particles (ExoM_LDL) significant reduction of this parameter in non-survivors when compared with survivors and the control group is traced both at the time of admission to the ICU (p<0.0001, p<0.0001, respectively) and 7 days after (p = 0.00043, p = 0.0089, respectively). Similar changes in the level of ExoM_LDL were observed on the first day of admission to the ICU but not in 7 days in the group of COVID-19 patients infected with the Alpha variant (S2b Fig, S2 Table).

## Survival predictive value of studied parameters by ROC analysis

ROC analysis was conducted to determine the threshold value of lipid concentrations (TC, LDL, HDL) measured by standard technique between survived and non-survived all patients with COVID-19, patients infected by the Alpha and the Delta variants separately. Significant threshold value was not found (data are not presented).

Also, ROC was performed to determine the threshold value of relative level of exomeres, LDL particles and conjoint fraction of exomeres and LDLs measured by DLS in patients with the Delta variants. Among all studied parameters reflecting lipid spectrum conjoint fraction of exomeres and LDL particles was the best to distinguish non-survivors and survivors at two estimated points (the time and 7 days after to admission to the ICU) (Fig 2). We have identified a threshold value for ExoM_LDL at the time of admission to the ICU as 0.272 (AUC = 0.903 (CI95%:0.793–1.00), specificity = 0.809, sensitivity = 0.889, accuracy = 0.846, p<0.0001) (Fig 2A). The threshold value for ExoM_LDL 7 days after the time of admission to the ICU was 0.284 (AUC = 0.909 (CI 95%: 0.773–1.000), specificity = 0.75, sensitivity = 1.00, accuracy = 0.80, p = 0.0007) (Fig 2B). Similar results have been demonstrated for patients with the Alpha variant. We have identified a threshold value for ExoM_LDL at the time of admission to the ICU as 0.296 (AUC = 0.796 (CI95%:0.616–0.976), specificity = 0.692, sensitivity = 0.800, accuracy = 0.750, p = 0.011) (S3H Fig).

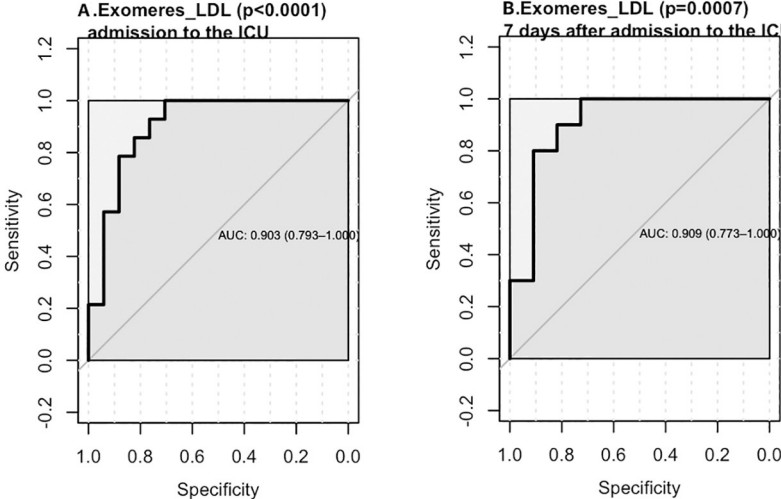

**Fig 2.** ROC analysis for analytes in blood plasma of patients with the Delta variant: A. ExoM_LDL (admission to the ICU), B. ExoM_LDL (7 days after admission to the ICU).

## Correlation analysis between lipid profile measured by DLS and standard technics

The matrices of correlation pairs of lipid profile and the ratio of the contributions to scattering parameters for exomeres (ExoM, ExoM_LDL, LDL) in the three above-mentioned groups (non-survivors with COVID-19, survivors with COVID-19, controls) in both time points for all COVID-19 patients and separately for COVID-19 patients infected by the Alpha and the Delta variants are presented in S3 and S4 Tables. We revealed a positive correlation between LDL measured by DLS and standard technique, namely, between LDL (rplU) with plasma LDL in survivors and non-survivors in both time points (at time to admission to the ICU and 7 days after admission to the ICU for all COVID-19 patients and separately for COVID-19 patients infected by the Alpha and the Delta variants (S3 and S4 Tables). Interesting, fraction ExoM_LDL was strong positive correlated with TC and LDL in survivors and non-survivors in all COVID-19 patients and separately for COVID-19 patients infected by the Alpha variant and only in survivors in group of COVID-19 patients infected by the Delta variant at time to admission to the ICU and in non-survivors in entire cohort of COVID-19 patients and separately for COVID-19 patients infected by the Alpha variant 7 days after to the ICU) (S3 and S4 Tables).

## Prediction of fatal outcome of COVID-19 using multivariable regression analysis

Multivariable regression analysis was performed to identify contribution of the parameters of lipid profile and exomeres to fatal outcome of all COVID-19 patients, patients infected by the Alpha and the Delta separately adjusted for hypertension, obesity, age and sex. We found the association between the higher relative level of conjoint fraction of exomeres and LDL particles (ExoM_LDL) and lower risk of fatal outcome of COVID-19 at the time of admission to the ICU (OR = 1.574e+21, CI95%:1.515e+09–1.635e+33, p = 0.0005; OR = 577.776, CI95%:4.473–74621.069, p = 0.0185, OR = 1.075, CI95%:1.032–1.120, p = 0.0015, respectively) (S5 Table).

## Validation of differential gene expression received in our transcriptomic data in PBMCs in the studied patients with COVID-19 with different outcome (survivor, non-survivor) infected by the Delta variant of SARS-CoV-2 at the time and 7 days after admission to the ICU and in the control group

We evaluated the expression levels of genes involved in cholesterol metabolism (*PPARG*, *LDLR*, *LRP6*, *CD36*, *STAB1*, *ANXA2*) in PBMCs of patients with various outcomes of COVID-19 (convalescence / death) infected by the Delta variant as variant associated with pronounce decrease of LDL level in non-survivors over time, as well as in controls (S6 Table). Studied genes were selected through our previous transcriptomic analysis that revealed upregulation of genes involved in *LDLR* activity pathway. At the time of admission to the ICU, increased *PPARG* and *CD36* gene expression levels were found both in non-survivors and survivors when compared with controls (*PPARG*: p = 0.00024, p = 0.00048, respectively; *CD36*: p = 0.0027, p<0.0001, respectively), as well as *ANXA2* gene expression level was higher only in survivors (p = 0.011) (S6 Table). After 7 days in the ICU increased *STAB1*, *CD36*, *LDLR* and simultaneously decreased *ANXA2* gene expression were detected in both non-survivors and survivors compared with the control group (p<0.05). Also, decreased expression levels of *PPARG* and *ANXA2* after 7 days treatment compared with the time of admission were revealed both in non-survivors and survivors (*PPARG*: p = 0.016, p = 0.016, respectively; *ANXA2*:

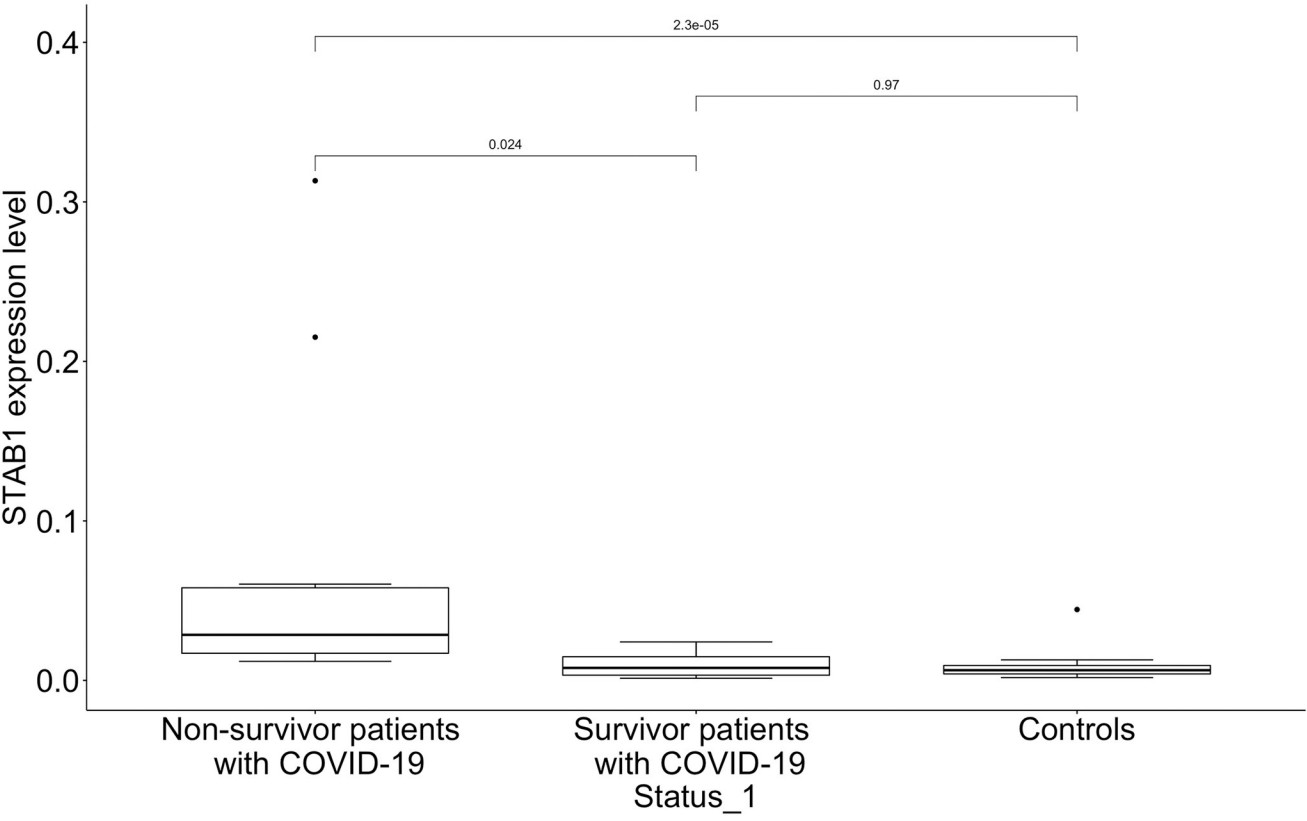

**Fig 3. *STAB1* expression level in PBMCs of COVID-19 patients infected by the Delta variant with different outcomes and controls.**

p = 0.00098, p = 0.0029, respectively). *LRP6* mRNA level was decreased in survivors when compared with the control group in 7 days after admission to the ICU (p = 0.0089). Additionally in survivors *LRP6* decreased 7 days after admission to the ICU when compared with the time to admission the ICU (p = 0.02).

It should be noted that only for the *STAB1* gene differential expression between non-survivors and survivors was demonstrated (p = 0.017) and this was specific only for 7 days after admission to the ICU. In non-survivors *STAB1* expression level was increased 7 days after admission to the ICU compared with the time of the admission (p = 0.0078). ddPCR was used for confirmation of this result. mRNA copy counts were determined for the *STAB1* gene in non-survived patients with COVID-19, survived patients with COVID-19 in 7 days after admission to the ICU and in control individuals. Increased expression of *STAB1* of non-survivors compared with survivors and controls was revealed (p = 0.024, p<0.0001, respectively). So, the result obtained by RT-PCR and demonstrating distinguishing differences between expression level of the *STAB1* gene in non-survivors and survivors was supported by ddPCR (Fig 3). However, ddPCR did not reveal the differences in expression level of the *STAB1* gene between controls and survived patients with COVID-19 (p>0.05) in 7 days after admission to the ICU.

## Discussion

Here, we for the first time carried out simultaneous assessment of the complex of parameters that determine cholesterol metabolism in order to find relevant biomarkers of poor prognosis

and fatal outcome of severe COVID-19. Plasma LDL and HDL levels were decreased in non-survivors infected by the Alpha or the Delta variants compared to controls at the time of admission to the ICU. However, the Delta variant has been associated with a more pronounce decrease in LDL levels in non-surviving patients during the time spent in the ICU. The conjoint fraction of exomeres and LDL particles measured by dynamic light scattering (DLS) was decreased in non-survivors infected by the Alpha and the Delta variants compared to survivors and controls at the time of admission to the ICU that may be considered as potential predictor of fatal outcome.

Over the past two years, link between cholesterol metabolism and severity of COVID-19 was shown. As, it has also been demonstrated that COVID-19 is accompanied by dyslipidemia [2, 3]. In particular, serum lipid levels decrease after the acute onset of COVID-19 and continue to decrease until the patient recovers [3]. Serum LDL levels have been found to be a predictor of poor disease prognosis of COVID-19 [4, 19]. Patients with COVID-19 with a fatal outcome were characterized by irreversible decrease of LDL reduced by up to 60% compared with the level at hospitalization [19]. However, this study operated on a small cohort of patients with COVID-19 which included 21 patients with COVID-19 with concomitant chronic pathologies, among them only 4 cases of death. Aparisi and colleagues also demonstrated the association of low LDL serum level with 30-days all-cause mortality in patients with COVID-19 [20]. This study included representative sample number (654 patients, 149 cases of death), but expected the patient group was heterogeneous with apparent prevalence of diabetes mellitus, dyslipidemia, as well as other comorbidities such as chronic kidney disease, ischemic heart disease, which were maximally excluded in our study. Our data generally confirmed these findings: LDL level was decreased 7 days after admission to the ICU in non-survivors compared to survivors independent of SARS-CoV-2 variants. It should be noting that proinflammatory cytokines induced by viral infection may modulate lipid metabolism [21]. Our study found increased secretion of pro-inflammatory cytokines such as CCL20/MIP3α, IL-10, IL-15, IL-27 in non-survivors than in controls that is consistent with previously published data (Supplementary materials) [22]. Also, our finding is that increased expression levels of LDL receptors genes (*LDLR*, *CD36*, *STAB1*) were found in non-survivors and survivors compared to controls at time to admission to the ICU. *LDLR* encodes canonical LDL receptor, *CD36* and *STAB1* encode important scavenger receptors for modified, primarily oxidized LDL (oxLDL). It was shown earlier that COVID-19 severity is associated with oxidative stress and products of lipid peroxidation accumulate during disease [23]. Thus, elevated *CD36* and *STAB1* gene expression may indicate increased oxidized lipoproteins uptake during severe COVID-19 and this in turn partially could explain lipid profile abnormalities observed in our study as well as in others. Survived patients also have increased *STAB1* gene expression compared with controls but significantly lower than in non-survivors 7 day after that supported the results of our previous transcriptome analysis [5]. In particular plasma TC, LDL and HDL cholesterol levels were markedly reduced in non-survivors when compared with the control group at the time of admission to the ICU and 7 days after. And while lipid concentrations tended to go up in survivors during treatment in the ICU they were continuously reduced in non-survivors in case of the Delta variant of SARS-CoV-2.

Increased activity of LDL receptors genes in our study reinforces data on enhanced lipid accumulation within cellular lipid droplets observed in SARS-CoV-2 infected cells, as well as in the lungs of deceased patients with COVID-19 [24]. Cholesterol is essential for coronavirus entry, membrane fusion, and pathological syncytia formation [25]. Interaction with ACE2 is facilitated by binding cholesterol to receptor-binding domain (RBD) of SARS-CoV-2 spike protein [26]. At the same time mutations in the RBD were linked to stability of S protein, increased ACE2 binding affinity and contagiousness of the Delta variant [27] therefore this

variant could more actively influence cholesterol metabolism and could lead to more pronounced decrease of lipid concentrations.

Additionally, to lipid profile we analyzed cholesterol containing particles in plasma primarily LDLs and extracellular vesicles by alternative method based on immunoadsorption combined with DLS. Our previous study linked production of exomeres to cellular cholesterol synthesis [13]. We and others believe now that exomeres are responsible for essential part of plasma cholesterol [13, 28]. Here we first assessed the onjoint level of particles with hydrodynamic radius of 20 nm (exomers and LDLs) as well as exomeres and LDLs fractions separately. Relative LDL level determined using this approach generally repeated LDL profiles in the studied groups.

At the same time, we first observed marked decrease in total proportion of exomeres and LDL particles in non-survivors independently of the Alpha or the Delta variants when compared with both survivors and the control group even at the time of admission to the ICU. The role of conjoint fraction of exomeres and LDL particles as predictor of fatal outcome of COVID-19 was also supported via multivariable regression analysis adjusted for hypertension, obesity, age and sex. Various opinions exist concerning specificity of decreased LDL level and COVID19 outcome. Different points of view suggests that a significant decrease in LDL cholesterol level is also a feature of mortality during sepsis [29] or, alternatively, ICU patients receiving membrane oxygenation but not lethality [30]. According to our data conjoin fraction of exomeres and LDL particles may be more accurate than simple estimation of plasma LDL cholesterol concentration at the time of admission to the ICU as a relevant predictor of fatal outcome. However, the nature of the decrease of these classes of particles quantity is unknown. Previous study showed that exomeres are enriched in ecto-domain fragments of ACE2 receptor and thus can bind to SARS-CoV-2 S protein S1 subunit [31]. Possible application of EVs and exomeres containing fully post-translationally modified ACE2 that can be used for an attenuation of SARS-CoV-2 infection ius being discussed [6, 31, 32]. The exact role of exomeres in SARS-CoV-2 infection needs further investigations.

## Conclusion

The current study confirmed that the severity of changes in lipid profile may correlate with the outcome of COVID-19. Overall, deceased patients with COVID-19, regardless of the variant with which they were infected (the Alpha, the Delta), were characterized by decreased LDL level at the time of admission. Decreased conjoint fraction of exomeres and LDL particles was identified as predictor of a fatal outcome of COVID-19. The Delta variant was associated with negative dynamics of HDL and LDL plasma levels during the disease and upregulation of LDLR pathway genes. Further studies are needed to analyze if this observation could be referred to other critical illnesses or is specific to COVID-19 infection.

## Supporting information

**S1 Fig.** Principle of particle count estimation by dynamic light scattering (DLS) combined with immunoadsorption: A. Analysis of unprocessed plasma, B. Analysis of plasma after CD9 positive particles (exosomes, exomeres) depletion, C. Analysis of plasma after HSP90 positive particles (exomeres) depletion, D. Analysis of plasma after apolipoprotein B100 positive particles (LDLs, VLDLs) depletion. Abbreviations on the figure: ApoB100—apolipoprotein B100, Dh–hydrodynamic diameter, HSA–human serum albumin, LDL–low density lipoproteins, VLDL–very low-density lipoproteins.
(TIF)

**S2 Fig.** Relative level of plasma exomeres with concentration of TC bound by LDL for patients with COVID-19 infected by Alpha variant: A. ExoM (admission to the ICU), B. ExoM_LDL (admission to the ICU), C. LDL (admission to the ICU), D. ExoM (7 days after admission to the ICU), E. ExoM_LDL (7 days after admission to the ICU), F. LDL (7 days after admission to the ICU).
(TIF)

**S3 Fig.** ROC analysis for in blood plasma for all patients with COVID-19 (A-F): A. ExoM (admission to the ICU), B. ExoM_LDL (admission to the ICU), C. LDL (admission to the ICU), D. ExoM (7 days after admission to the ICU), E. ExoM_LDL (7 days after admission to the ICU), F. LDL (7 days after admission to the ICU); for patients with COVID-19 infected by the Alpha variant (G-L): G. ExoM (admission to the ICU), H. ExoM_LDL (admission to the ICU), I. LDL (admission to the ICU), J. ExoM (7 days after admission to the ICU), K. ExoM_LDL (7 days after admission to the ICU), L. LDL (7 days after admission to the ICU); for patients with COVID-19 infected by the Delta variant (M-Q): M. ExoM (admission to the ICU), N. LDL (admission to the ICU), O. LDL (admission to the ICU), P. ExoM (7 days after admission to the ICU), Q. ExoM_LDL (7 days after admission to the ICU), R. LDL (7 days after admission to the ICU).
(TIF)

**S1 File. Plasma cytokine profile in the studied patients with COVID-19 with different outcome (survivor, non-survivor) infected by the Delta variant of SARS-CoV-2 at the time to admission to the ICU and in the control group.**
(PDF)

**S1 Table. Primers and probe sequences for estimation expression levels of the studied genes.**
(DOCX)

**S2 Table. Relative level of plasma exomeres for patients with COVID-19 with concentration of TC bound by LDL.**
(DOCX)

**S3 Table. Correlation analysis between lipid profile parameters and SIC exomeres in patients with COVID-19 at the time of admission to ICU and in control group.**
(DOCX)

**S4 Table. Correlation analysis between lipid profile parameters and SIC exomeres in patients with COVID-19 7 days after admission to the ICU.**
(DOCX)

**S5 Table. The association between studied parameters of lipid parameters measured by DLS and standard technics for patients with COVID-19.**
(DOCX)

**S6 Table. mRNA levels of the studied genes in PBMCs of patients with severe COVID-19 with Delta variant and in the control group.**
(DOCX)

## Author Contributions

**Conceptualization:** Sofya Pchelina.

**Data curation:** Sofya Pchelina.

**Formal analysis:** Tatiana Usenko, Anastasia Bezrukova.

**Investigation:** Tatiana Usenko, Anastasia Bezrukova, Katerina Basharova, Sergey Landa, Zoia Korobova, Natalia Liubimova, Ivan Vlasov, Mikhael Nikolaev, Artem Izyumchenko, Elena Gavrilova, Irina Shlyk, Elena Chernitskaya, Yurii Kovalchuk.

**Methodology:** Tatiana Usenko.

**Project administration:** Sofya Pchelina.

**Software:** Tatiana Usenko.

**Supervision:** Sofya Pchelina.

**Visualization:** Tatiana Usenko, Valentina Miroshnikova, Katerina Basharova.

**Writing – original draft:** Tatiana Usenko.

**Writing – review & editing:** Valentina Miroshnikova, Petr Slominsky, Areg Totolian, Yurii Polushin, Sofya Pchelina.

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
