## [Decision Letter · Decision Letter 0]

19 Dec 2022

PONE-D-22-30828The role of cholesterol metabolism in predicting clinical outcome of patients with severe COVID-19PLOS ONE

Dear Dr. Usenko,

Thank you for submitting your manuscript to PLOS ONE. After careful consideration, we feel that it has merit but does not fully meet PLOS ONE’s publication criteria as it currently stands. Therefore, we invite you to submit a revised version of the manuscript that addresses the points raised during the review process. Please carefully address the comments from Reviewer 2. If you need longer time for revision, please contact PLOS ONE.

We look forward to receiving your revised manuscript.

Kind regards,

Yi Cao

Academic Editor

PLOS ONE

Journal Requirements:

"This work was supported by the Genome Research Centre development program «Kurchatov Genome Centre» (agreement No.075-15-2019-1663)"

5. Please remove your figures from within your manuscript file, leaving only the individual TIFF/EPS image files, uploaded separately. These will be automatically included in the reviewers’ PDF. 

6. Please upload a new copy of Supplementary Figure 4 as the detail is not clear. Please follow the link for more information:

https://blogs.plos.org/plos/2019/06/looking-good-tips-for-creating-your-plos-figures-graphics/

https://blogs.plos.org/plos/2019/06/looking-good-tips-for-creating-your-plos-figures-graphics/

Reviewers' comments:

Reviewer's Responses to Questions

**Comments to the Author**

1. Is the manuscript technically sound, and do the data support the conclusions?

Reviewer #1: Yes

Reviewer #2: No

2. Has the statistical analysis been performed appropriately and rigorously? 

Reviewer #1: Yes

Reviewer #2: No

3. Have the authors made all data underlying the findings in their manuscript fully available?

Reviewer #1: Yes

Reviewer #2: Yes

4. Is the manuscript presented in an intelligible fashion and written in standard English?

Reviewer #1: No

Reviewer #2: No

5. Review Comments to the Author

Reviewer #1: 1. The introduction contains parts of the method section. Only relevant information should be presented in the introduction.

2. Have patients' medications been considered? lipid-lowering agents, especially PCSK9 inhibitors.

3. Why was the evaluation of the patients made after 7 days?

4. How was the number of individuals enrolled in the study determined? Was the sample size calculated?

Reviewer #2: In this study, the authors assessed the relationship between cholesterol metabolism and clinical outcome of patients with severe COVID-19. They show that a downward trend of cholesterol is associated with poorer prognosis.

The title of the paper suggests that the study is about cholesterol metabolism and outcome, but we are also given lots of information about cytokine levels and the correlation between lipid profiles measured with DLS and standard techniques. All this moves the attention a bit away from the main message: downregulation of cholesterol and prognosis. The authors need a lot of space to make their point and mix a lot of data. This makes the paper difficult to read. Why not describing the relationship between cholesterol fractions and prognosis (with more patients, of course) as a clinically relevant observation. All the other things are more mechanistic in nature and will not immediately interest the practicing physician.

The paper needs considerable language polishing.

Introduction

This part of the paper is unduly long and not so easy to read. It is not focused well on the problem which needs to be studied. It also lacks a clear hypothesis.

Methods

The authors should clarify the approval by the Ethics Board in more detail. The present study is a retrospective analysis, but all participants apparently signed a written consent prior to the study. How is that possible when patients died? This is particularly relevant with respect to the replication cohort.

Were all patients admitted in July and August included in the study or was there some selection?

The replication study included patients had had been admitted several months before the main study. This is odd because a replication study usually follows the main study and needs to have a prospective design. Here, the replication study is a retrospective analysis as well with all its inherent shortcomings and can, therefore, not truly serve as a replication.

6. PLOS authors have the option to publish the peer review history of their article (what does this mean?). If published, this will include your full peer review and any attached files.

Reviewer #1: **Yes: **Medine Cumhur Cure

Reviewer #2: **Yes: **Peter W de Leeuw

---

## [Author Response · Author response to Decision Letter 0]

24 Jan 2023

Dear reviewers, we greatly appreciate all comments.

Reviewer #1: 

1. The introduction contains parts of the method section. Only relevant information should be presented in the introduction. 

- Was shortened and corrected.

2. Have patients' medications been considered? lipid-lowering agents, especially PCSK9 inhibitors. 

- As the inclusion criteria include the absence of chronic comorbidities such as cancer, cerebrovascular diseases, heart failure, or renal failure all patients included in the study did not receive the lipid-lowering therapy. This information was included in the text.

3. Why was the evaluation of the patients made after 7 days? 

This day was chosen as the first point suitable for assessing the effectiveness of pathogenic therapy. The average time a patient spends in the intensive care unit of Pavlov First Saint-Petersburg State Medical University is 7 days.

4. How was the number of individuals enrolled in the study determined? Was the sample size calculated? 

All patients admitted to the intensive care unit and met the inclusion criteria were included in the study. Blood samples were collected in two waves, during an infection with alpha and delta variants of SARS-CoV-2 virus. This information was added in the text.

Reviewer #2: In this study, the authors assessed the relationship between cholesterol metabolism and clinical outcome of patients with severe COVID-19. They show that a downward trend of cholesterol is associated with poorer prognosis.

The title of the paper suggests that the study is about cholesterol metabolism and outcome, but we are also given lots of information about cytokine levels and the correlation between lipid profiles measured with DLS and standard techniques. All this moves the attention a bit away from the main message: downregulation of cholesterol and prognosis. The authors need a lot of space to make their point and mix a lot of data. This makes the paper difficult to read. Why not describing the relationship between cholesterol fractions and prognosis (with more patients, of course) as a clinically relevant observation. All the other things are more mechanistic in nature and will not immediately interest the practicing physician. 

Thank you very much. The title of the article has been changed and the main body of the article restructured in order to strengthen the link of blood cholesterol fractions and COVID-19 prognosis.

The paper needs considerable language polishing.

Introduction

This part of the paper is unduly long and not so easy to read. It is not focused well on the problem which needs to be studied. It also lacks a clear hypothesis. 

-The Introduction section was shortened and the following supposition was added “We suppose that exomers may play a role in COVID-19 pathogenesis as an essential role of exomeres in cholesterol transport was shown. To further analyze the role of cholesterol metabolism in poor prognosis of COVID-19 we evaluate retrospectively the parameters of cholesterol metabolism (plasma lipids concentrations and exomere fraction, the expression levels of LDLR activity pathway genes selected through our previous transcriptome analysis) as well as cytokine profile as possible predictors of fatal outcome in patients with severe COVID-19 infected by alpha and delta variants of SARS-CoV-2 virus”.

Methods

The authors should clarify the approval by the Ethics Board in more detail. The present study is a retrospective analysis, but all participants apparently signed a written consent prior to the study. How is that possible when patients died? This is particularly relevant with respect to the replication cohort. 

-The written concern was signed in the time of admission in the intensive care unit.

Were all patients admitted in July and August included in the study or was there some selection? 

-All patients admitted to the intensive care unit and met the inclusion criteria were included in the study. This information was added in the text.

The replication study included patients had had been admitted several months before the main study. This is odd because a replication study usually follows the main study and needs to have a prospective design. Here, the replication study is a retrospective analysis as well with all its inherent shortcomings and can, therefore, not truly serve as a replication. 

-We made a correction and aggregated the study in one with a comparison between patients with COVID-19 infected by еру Alpha and Delta variants of SARS-CoV-2 virus. It allows us to conduct a comparison between analyzed marker for poor COVID-19 prognosis among patients affected with different variants of SARS-CoV-2 virus. Particularly, an information about an association of the Delta variant with a more pronounce decrease in LDL levels in non-surviving patients during the time spent in the ICU is discussed.

---

## [Decision Letter · Decision Letter 1]

30 Jan 2023

Fraction of plasma exomeres and low-density lipoprotein cholesterol as a predictor of fatal outcome of COVID-19

PONE-D-22-30828R1

Dear Dr. Usenko,

We’re pleased to inform you that your manuscript has been judged scientifically suitable for publication and will be formally accepted for publication once it meets all outstanding technical requirements.

Kind regards,

Yi Cao

Academic Editor

PLOS ONE

Additional Editor Comments (optional):

Reviewers' comments:

Reviewer's Responses to Questions

**Comments to the Author**

1. If the authors have adequately addressed your comments raised in a previous round of review and you feel that this manuscript is now acceptable for publication, you may indicate that here to bypass the “Comments to the Author” section, enter your conflict of interest statement in the “Confidential to Editor” section, and submit your "Accept" recommendation.

Reviewer #1: All comments have been addressed

2. Is the manuscript technically sound, and do the data support the conclusions?

Reviewer #1: Yes

3. Has the statistical analysis been performed appropriately and rigorously? 

Reviewer #1: Yes

4. Have the authors made all data underlying the findings in their manuscript fully available?

Reviewer #1: Yes

5. Is the manuscript presented in an intelligible fashion and written in standard English?

Reviewer #1: Yes

6. Review Comments to the Author

Reviewer #1: I have no further comments. The authors improved the article according to the recommendations of the reviewers.

7. PLOS authors have the option to publish the peer review history of their article (what does this mean?). If published, this will include your full peer review and any attached files.

Reviewer #1: **Yes: **Medine Cumhur Cure

---

## [Editor Report · Acceptance letter]

1 Feb 2023

PONE-D-22-30828R1 

Fraction of plasma exomeres and low-density lipoprotein cholesterol as a predictor of fatal outcome of COVID-19 

Dear Dr. Usenko:

I'm pleased to inform you that your manuscript has been deemed suitable for publication in PLOS ONE. Congratulations! Your manuscript is now with our production department. 

Kind regards, 

on behalf of

Dr. Yi Cao 

Academic Editor

PLOS ONE